# Epilepsy and Diagnostic Dilemmas: The Role of Language and Speech-Related Seizures

**DOI:** 10.3390/jpm12040647

**Published:** 2022-04-18

**Authors:** Soultana Papadopoulou, Efterpi Pavlidou, Georgios Argyris, Thaleia Flouda, Panagiota Koukoutsidi, Konstantinos Krikonis, Sidrah Shah, Dana Chirosca-Vasileiou, Stergios Boussios

**Affiliations:** 1Department of Speech and Language Therapy, University Hospital of Ioannina, 45111 Ioannina, Greece; soultpap@yahoo.gr (S.P.); efterpi.pavlidou@gmail.com (E.P.); 2ENT Private Medical Office, 45444 Ioannina, Greece; argyrisgiwrgos@gmail.com; 3Linguist Private Practice, 45332 Ioannina, Greece; thaleia.flouda@yahoo.gr; 4School of Medicine, Biomedical Engineering, University of Patras, 26331 Patras, Greece; drosfp@gmail.com; 5Statistics and Research Design Company, DatAnalysis, 45221 Ioannina, Greece; krikonis@yahoo.com; 6Department of Palliative Care, Guy’s and St Thomas’ Hospital, Great Maze Pond, London SE1 9RT, UK; sidrah.shah@nhs.net; 7Department of Neurology, Medway NHS Foundation Trust, Windmill Road, Gillingham ME7 5NY, UK; d.chirosca@nhs.net; 8Department of Medical Oncology, Medway NHS Foundation Trust, Windmill Road, Gillingham ME7 5NY, UK; 9Faculty of Life Sciences & Medicine, School of Cancer & Pharmaceutical Sciences, King’s College London, London SE1 9RT, UK; 10AELIA Organization, 9th Km Thessaloniki-Thermi, 57001 Thessaloniki, Greece

**Keywords:** epilepsy, seizures, language-induced epilepsy, stuttering, psychogenic nonepileptic seizures

## Abstract

Although the impact of epilepsy on expressive language is heavily discussed, researched, and scientifically grounded, a limited volume of research points in the opposite direction. What about the causal relationship between disorder-related language activities and epileptic seizures? What are the possible diagnostic dilemmas that experts in the field of speech-language pathology, neurology, and related fields face? How far has research gone in investigating psychogenic nonepileptic seizures, the misdiagnosis of which can be a thorny issue for clinicians and a detrimental factor for the patients’ health? In order to address these questions, the study at hand focuses on a common, ever-intensified (by the COVID-19 pandemic) speech disorder—stuttering, and explores the pathophysiological and psychogenic background of the phenomenon. It also looks at the role of stuttering as a contributing factor to the appearance of epileptic seizures, in the hope of drawing attention to the complexity and importance of precise detection of stuttering-induced epilepsy, as a specific subcategory of language-induced epilepsy.

## 1. Introduction

It has been widely proven that epilepsy can affect language. Admittedly, the type, severity, and main cause of epilepsy, along with the type of treatment, define the extent and nature of the derived language disturbance (Figure 1) [1]. It has been specifically reported through case studies that certain antiepileptic drugs, such as phenytoin, carbamezepine, lamotrigine, topiramate, valproate and levetiracetam gabapentin, and divalporoex sodium have either induced the appearance or prevented the onset of language dysfunctioning, depending on the patients’ clinical background and drug idiosyncrasy, thus, strengthening the initial hypothesis of this work [2,3]. Similarly, one of the studies reviewed reports that ethosuximide and phenobarbital therapy prescribed to a 47 year old male patient with epileptic seizures reduced clinical seizures, including stuttering, while no receipt of anticonvulsant medication during certain periods revealed that clinical seizures were repeatedly percipitated by certain stimuli [4]. As presented and proven below, a great volume of literature investigates the link between epileptic seizures and perforce disturbances in language and speech fluency—including stuttering, in particular. Yet, a significant low number of studies looks at how stuttering may trigger epileptic seizures. The ultimate objective of this manuscript is to observe any trends in the perspectives through which published articles investigate the topic, and to shed light on the relevant diagnostic dilemma.

## 2. Background and Methods

A disanalogous examination of the topic may intensify diagnostic errors in the field of epileptology. A limited volume of research points in the opposite direction, contributing, thus, less to the actual diagnostic and prevention issues experts face in speech-language pathology, neurology, and related fields. The scientific community still has a lot to cover on epileptic seizures, language disturbances, and the relationship between the two phenomena. The main questions addressed in this work are: “what about the causal relationship between disorder-related language activities and epileptic seizures?”, “to what extent do case studies and literature reviews examine the role of speech- fluency disturbances in the triggering process of certain types of epileptic seizures?”, and finally, “how far has research gone in investigating pseudoseizures, the misdiagnosis of which can be a thorny issue for the clinicians and a detrimental factor for the patient’s health?”.

More specifically, an extensive literature review, performed on Medline, Embase, Cohrane, and Pascal databases with articles published on PubMed and Scopus in the period 1960 and January 2022 reveals that the initial hypothesis on the disanalogous investigation of the topic holds true. In fact, the greatest volume of literature investigates how epilepsy affects language. Interestingly, the number of publications on the topic drops to 50% when it comes to scientific articles on language-induced epilepsy, and only 62 articles engage in a study of how stuttering and epilepsy are connected (see Table 1 & Figure 2 below).

The literature review has been conducted with the keywords: epilepsy affects language, language induced epilepsy, stuttering and epilepsy, epileptic seizures misdiagnosis. The research was limited to humans and the English language, while a detailed examination of published articles was conducted for the period between 1960 and 2022, clearly during the COVID-19 pandemic measures, to assess how reflex epilepsy and/or stressed-induced stuttering has been investigated as a possible factor triggering epileptic spells. The final selection of articles studied was conducted according to which of these articles present cases or provide literature reviews on the topics of stuttering and language induced epileptic seizures, in order to provide insight as the complexity of the phenomenon, its relation to epilepsy and literature trends.

## 3. Why Stuttering?

Stuttering or stammering is a common speech disorder. It is a vocal phenomenon characterized by disturbances in the flow of speech and according to DSM-V-TR (Diagnostic and Statistical Manual of Mental Disorders, 5th Edition, Text Revision) specifically, by the prevalence of at least one of the following: broken words, sound and syllable repetitions, prolongation of sounds, interjections, audible or silent blocking, circumlocutions, and monosyllabic whole word repetitions [1,5,6].

The severity of stuttering may vary significantly among individuals—depending on different factors, including communication circumstances, fatigue, and anxiety levels. The phenomenon is of increasing interest to researchers in different fields, including speech and language pathology, neurology, genetics, linguistics, and neurosciences, as more than 70 million people worldwide stutter, according to the Stuttering Foundation. Apart from what numbers say, however, it is challenging to reach conclusions as to why stuttering is a point of debate and/or investigation, as it is intricately interwoven and connected with a great number of neurological and psychogenic disorders. Yet, crudely put, stuttering is common in young children (an expected occurrence as part of the process of speech development between ages 3 and 8), a chronic condition occasionally seen to persist into adulthood, a symptom of trauma, brain injury, strokes, or of epileptic seizures [7,8]. It can be a sign of neurodegenerative disease, as well as a symptom of psychogenic background, related to mental disorders or social anxiety. It is important to report here that two of the reviewed studies, conducted in 2009 and 2013 among 73 children and 92 adults with stuttering issues, draw a link between stuttering and social phobia, generalized anxiety disorder, panic disorder, and social anxiety disorder [9,10].

What is interesting, however, is that stuttering as a fluency disorder is heavily seen as a symptom rather than a diagnosis. One reason to explain this one-sided perspective in the literature is that the neurophysiology of stuttering is still unclear, as it is considered a complex process [8]. Generally, the idea that the majority of people use one cerebral hemisphere, usually the left, for speech, but certain individuals, commonly left-handed or ambidextrous ones, organize their speech mechanisms in both cerebral hemispheres, is still investigated [11]. Another reason is that the onset of stuttering in adulthood, with no prior history, family background, or demonstrable neurological insult, makes a possible diagnosis less profound [12]. Moreover, the fact that lately, the literature on the topic at hand focuses on the possible connection of stuttering and other neurological problems with COVID-19, further obstructs the diagnostic parameters of stuttering, as one case study reports that there is a connection between a 53 year-old female patient’s stuttering and word-finding difficulties were due to a response to the COVID-19 infection [13]. Another study conducted in the United Kingdom within 2020 presents the first nationwide, cross-specialty surveillance study of acute neurological and psychiatric complications of COVID-19, and although it does not directly link COVID-19 with stuttering, it draws attention to unexpected complications, and paves the way for further research [14].

As the aforementioned perspectives–which observe stuttering as a symptom, mainly in epilepsy–are heavily discussed in published articles from 1960’s until today and the latter, COVID-19 outlook is still an emerging vast research field, and this study looks at language-induced epilepsy. Acquired stuttering is a manifestation of language-induced epileptic seizures—the examination of the phychogenic and neurological background of which, can shed more light as to how stuttering is connected to reflex and language-induced epilepsy. At large, such a study, can draw attention to a further investigation of the diagnostic dilemma of whether stuttering, reflex epilepsy, and language-induced epilepsy are the established effect or the questioned cause.

At this point it is imperative that reflex epilepsy, language-induced epilepsy, and any possible connections with stuttering are briefly explained—for purposes of flow and clarity, rather than for further investigation.

## 4. On Reflex Epilepsy

Reflex is a carefully selected term to describe those instances in which a highly patterned stimulus or movement regularly leads to a seizure. This is in contrast to other forms of epilepsy, in which the seizure-provoking agent may be a local metabolic change within the focus, or a change in some blood-borne substance capable of influencing the focus [15]. The frequency of reflex epilepsies depends on the type, and can reach as high as 25% for photosensitive epilepsy, television-induced epilepsy, or video-game induced epilepsy, as alternatively termed [16].

Although more types of reflex epilepsy, such as musicogenic, eye-closure, orofacial reflex, myoclonic, and praxis induction epilepsy have been defined, two possible major types of reflex epilepsy, termed primary or idiopathic reflex epilepsy and secondary or symptomatic reflex epilepsy, respectively, have been observed and classified by the International League Against Epilepsy (ILAE) Task Force on Classification and Terminology Organization. In cases of primary or idiopathic reflex epilepsy, cortical foci are the rule, and a high association with family history and early-life appearance have been observed [1,17]. In secondary or symptomatic reflex epilepsy, on the other hand, a highly specific class of stimuli is effective, although the actual individual stimulus may be different in each instance and an occurrence later in life in patients with associated neurological and non-epileptic impairment [5,18]. With regards to prognosis and treatment, primary reflex epilepsy is usually benign with good response to medication [19,20], while symptomatic reflex epilepsy does not present standard symptomatology, and response to drugs for focal seizures triggered by specific stimuli is quite poor [21].

## 5. On Language Induced Epilepsy

Language-induced epilepsy is a subcategory of reflex epilepsy during which specific language stimuli appear to be the triggering mechanism. Specifically, higher mental activities, such as reading, speaking, writing, calculating, concentrating, playing chess, reading music, and playing a musical instrument, among others, have been reported as triggering focal or generalized seizures, under certain circumstances. To avoid misconceptions, it is deemed important here to exclude seizures triggered by non-verbal higher brain activities related to spatial processing and ideation or movements from the category of language-induced epilepsy, as such are considered praxis-induced seizures.

Language in any of the three modalities—reading, writing and speaking—has been reported in our literature review as a seizure-provoking stimulus [22]. This type of epilepsy is used to describe seizures provoked by failed attempts to speak, read, or write, while the phenomenon–although associated with inextricable facets in patients’ daily routine–is only partly investigated in published scientific works [3]. The literature review conducted for the purposes of the present work has also revealed that there is limited investigation on graphogenic or writing epilepsy as another variant of language induced epilepsy [23]. Yet, the existence of such studies points to the need of further investigating how some tasks involving complex mental involvement for activities performed by the hands confirm the precipitation of myoclonic jerks in patients with juvenile myoclonic epilepsy [23,24]. Admittedly, writing is an intensive mental activity which involves praxis sub-activity, an observation which not only explains the different categorizations between general praxis-induced epilepsy and graphogenic epilepsy, but also emphasizes the causal relationship between stimulus and the emergence of epileptic seizures.

In a similar way, our research conveys that there are reported cases of patients with seizures having occurred upon an attempt to speak [12,21]. Language-related tasks, including reading, can induce seizures, while many cases of language-induced epilepsy, caused by argumentative talking and writing or even singing and recitation, have been reported to a greater extent (compared to reading triggered epilepsy), pinpointing language-induced epilepsy as less debated by publications [25,26].

## 6. The Role of Stuttering and Connection with Reflex-Epilepsy

The connection between stuttering and stress has been the subject of debate in the fields of speech therapy and mental health for years, and the prevailing theory throughout the 20th century has accepted psychological factors, such as stress, rather than physiological ones, as possible causes for stuttering to take place. Case studies and other types of research conducted over the years have linked stuttering as a phenomenon of social anxiety for adolescents and adults, with a study in 2009 publishing that 50% of adults who stutter have social anxiety. Specifically, Dr. Lisa Iverach’s studies in 2009 and 2014 make a point that stuttering can be a direct cause for social anxiety, rather than a side-effect, and that speech therapy intervention is needed to correct stuttering, among other interventions targeting possible psychogenic or neurophysiological foundations of the phenomenon.

However, stuttering has also been associated with seizures, especially under stressful periods in patients’ lives [27]. The notion that stuttering is “a relative of epilepsy” over the past decade is further intensified by studies reporting the possibility that some relationship might exist between speech impediment and epilepsy. Other studies directly suggest stuttering as the cause and the epileptic crisis as the result, while a study review shows that rates of stuttering among patients with epilepsy are higher than in the general population [28]. Moreover, it seems that epilepsy is more frequently encountered among children with stuttering compared to children without stuttering—findings which indicate a link between stuttering and epilepsy [29,30].

The link was also evident in a case study published in 1988, reporting cases in which patients occasionally stutter during spontaneous reflex seizures, and that stuttering may even be the only manifestation of the episode. Remarkably, the phenomenon of stuttering being completely vanished after a recovery from a seizure strengthens the previously supported hypothesis that stuttering may be the cause and not necessarily the outcome of the seizure [31].

Other scientists remark that speech impediment appears to have been closely linked to the abnormal bioelectrical activity of the right temporal cortex. They believe that this abnormality may be the factor that triggers stuttering, and simultaneously is the factor that leads to epilepsy. Therefore, they support that epilepsy and stuttering seem to be causally related [28]. Also, it has been proven that following brain trauma or brain intoxication (e.g., with copper), adults may present with stuttering and epilepsy, though not always simultaneously at the same period, so this could also suggest that the cause of both epilepsy and stuttering is the same [32]. In a classic case study, a person who had chronic epilepsy and stuttering underwent neurosurgery under local anesthesia, and during this operation the patient started to speak fluently. This improvement remained consistent post-operatively, and epileptic seizures vanished simultaneously.

This case indicated that perhaps the origin is the same in both stuttering and epilepsy. Indeed, according to many researchers, this link between stuttering and epilepsy is clearly causal.

A family study involved nine members of three generations with Language-Induced Epilepsy, Acquired Stuttering, or Idiopathic Generalized Epilepsy. All patients underwent video-polygraphic electroencephalogram (EEG) recordings both sleeping and awake. The study demonstrated the phenotypic heterogeneity of the association of Idiopathic Generalized Epilepsy phenotype with Ictal Stuttering (Language-Related Reflex Seizure), and suggested that this form of Reflex Epilepsy related to language has more similarities with generalized epilepsy than with focal ones.

## 7. Pathophysiology and Genetics of Stuttering—A Bridge to Epilepsy

As previously mentioned, a link between stuttering and epilepsy is suggested by bibliography, but the nature of the link is yet to be clarified. Given the high prevalence and the severity of the disease, the pathophysiological mechanisms of human epilepsy are well-studied, while the pathophysiology of stuttering remains still obscure. Modern brain-activity recording methods and neuroimaging techniques attempt to provide insight into the mechanisms underlying the clinical manifestation of stuttering.

Some of the first studies suggested that there is incomplete lateralization or abnormal cerebral dominance in people who stutter, with the cerebral hemispheres holding opposing roles. The left hemisphere is considered related to the production of stuttered speech, while the right one may act in a compensatory manner to the symptom [33]. In 2000, Salmelin and colleagues used whole-head magnetoencephalography (MEG) in developmental stutterers (DS) and fluent speakers in an attempt to record the sequence of cortical activation, while subjects read aloud and vocalized single words. DS presented cortical activation first in the motor cortex and premotor area (associated with motor programming), and immediately after to the left inferior frontal region (associated with articulation and language processes). Fluent speakers exhibited the reverse pattern, and thus it seemed that DS initiates motor programs before articular code is prepared [34].

A neuroimaging study from Michigan State University in 2015 measured the fractional anisotropy derived from cerebral white matter using Magnetic Resonance Imaging (MRI) in children who stutter, and compared the respective measurements from fluent age-matched controls in an attempt to detect neuroanatomical differences. Scientists observed reduced fractional anisotropy in stuttering children relative to controls in white matter tracts that interconnect auditory and motor structures, in the corpus callosum, and in tracts interconnecting cortical and subcortical areas, which suggests possible structural connectivity deficits in this study group, a finding consistent with those of previous studies [35,36]. Another brain MRI study detected and compared regional Cerebral Blood Flow (rCBF) in a group of stutterers and a control group of fluent speakers. The study revealed decreased rCBF in Broca’s area (key component to speech production) and increased rCBF in cerebellar nuclei and parietal cortex (a possible compensatory mechanism) in the stuttering group compared to controls [36].

A most recent study of a large family with inherited stuttering, using T_1_-weighted and diffusion-weighted MRI, demonstrated a disruption in the cortico-basal ganglia-thalamo-cortical network (fundamental brain network in many activities, including initiating speech motor programs), an increase in globus pallidi bilaterally, and structural differences in Broca’s area between the study group and control group [37,38].

The implication of neurotransmitters was also considered. Maguire et al. studied a small group of DS before and after their treatment with risperidone (a D_2_/5-HT_2_ antagonist) using positron emission tomography (PET). In the risperidone-treated group, increased metabolism in the left striatum (caudate and putamen) and Broca’s area was observed, a finding that strengthens previous research that implicated the role of increased dopamine and striatal hypometabolism in stuttering [39].

Functional MRI (fMRI) became available after PET and nowadays dominates neuroimaging of stuttering due to its high spatial resolution. Speech production and resting state fMRI studies have reported several abnormalities in widely distributed brain regions, as well as in connectivity between regions of critical importance for speech organization and production [40]. However, fMRI studies appeared to have several limitations until nowadays [36,40].

The genetic basis of stuttering is still to be defined. Twin studies suggest that monozygotic twins display stuttering in higher rates compared to dizygotic twins, indicating a relatively strong genetic component, while studies in families with stuttering members attempt to identify a mode of inheritance [37,41]. Despite limitations carried by genomic studies, mutations in *GNPTAB* gene (encodes the enzyme N-acetylglucosamine-1-phosphotransferase) found in Pakistani families with stuttering, mutations in *AP4E1* gene (encoding adaptor protein complex 4) in a large Cameroonian study, and loci on chromosomes 1 and 4, determined by genetic mapping, in a large family with inherited stuttering, suggest an autosomal dominant pattern. Nevertheless, stuttering seems to be a complex trait, and more in-depth genetic research will improve current understanding of this clinical manifestation [37,41,42].

As indicated in previous sections of the current review and by relevant bibliography, the rates of stuttering among patients with epilepsy are higher than in the general population. The interrelation between epilepsy and stuttering is not straightforward, especially when stuttering is considered the stimuli for an epileptic seizure to occur, and not the clinical symptom of an epileptic seizure [29,30,43]. From a genetic perspective, intragenic deletions of the contactin-associated protein-like 2 gene (*CNTNAP2*) have been found in patients with epileptic syndrome and stuttering. Normally, the gene products are responsible for bridging the intercellular space between neurons. The *CNTNAP2* alleles that express the aforementioned deletions interfere with the physiological process of connecting neuronal cells and present a molecular basis for several neurodevelopmental disorders, including epilepsy and stuttering. Other conditions that are associated with intragenic deletions of the *CNTNAP2* are Gilles de la Tourette syndrome, intellectual disability, obsessive-compulsive disorder, language impairments, and attention deficit hyperactivity disorder [44].

Despite the limited available bibliography on epileptic seizures induced by language, there have been case reports and studies that used video-EEG with electromyogram (EMG) recordings in an attempt investigate stuttering patients. In a case of language-induced epilepsy, the patient exhibited facial myoclonus while reading aloud, and dysfluent language, mimicking stuttering. Paroxysmal discharges in EEG recordings of the left frontal region were consistently associated with a brief interruption of language. In this particular patient, silent reading did not induce any epileptic discharge, and as a result, articulatory movements during phonation were assumed to be the triggering factor. The patient was treated with antiepileptic medication [29]. Michel et al. conducted video-polygraphic EEG recordings in four patients with a diagnosis of juvenile myoclonic epilepsy (JME) in whom coexisted praxis- and language-induced jerks. Complex stimuli-reading and praxis-induced reflex seizures in these patients, characterized by facial myoclonias and stuttering. EEG recordings were indicative of brief paroxysms of very fast spikes followed by a slow wave, mainly in the frontocentroparietal areas [27]. In a neurophysiological study of nine members of a family with history of idiopathic generalized epilepsy (IGE) with interictal stuttering, spontaneous language was the main triggering factor for the occurrence of myoclonic jerks in five members. They also reported acquired stuttering (which was proved to be of epileptogenic origin). Stuttering was also present while reading, and EEG recordings exhibited abnormalities (e.g., spikes followed by slow waves) [21]. The above observations suggest that some forms of acquired stuttering could be linked to epileptic seizure, and electrophysiological studies may prove useful in investigating them. Although it was not possible to identify specific pathophysiological mechanisms shared by epilepsy and language as a trigger for epileptic seizure from the current bibliographical review, nor from the systematic literature that investigates both of them using modern neuroimaging techniques, it is apparent that there is ground for research to be covered.

## 8. Pharmacologic Implications

Several studies have highlighted stuttering as a side effect of anticonvulsant medication, an observation that concerns clinicians. A study conducted by Karimzadeh et al., testing the antiepileptic drug Zonisamide in children with refractory epilepsy, reported stuttering as a minor side effect of the drug, affecting 4.9% of participants [45]. Additionally, another antiepileptic medication used for seizure prophylaxis-phenytoin-induced in rare cases stuttering symptoms, which disappeared after drug discontinuation [43,46]. Bibliography also reports that stuttering may follow the administration of clozapine, an atypical antipsychotic medication [47].

On the contrary, anticonvulsant medication seems to improve stuttering symptoms in some cases of epilepsy. By reducing or even eliminating abnormal paroxysmal activity in the epileptic brain, there is improvement of neocortical functioning, and consequently, of fluency [3,43].

Clinicians should be aware of the possible, although rare, side effects of anticonvulsant medication as well as their potentially beneficial effect on speech impediment.

## 9. Psychogenic Nonepileptic Seizures vs. Epileptic Seizures—The Role of Ictal Stuttering

Among patients who are evaluated for refractory epileptic seizures, approximately 25% are found to have psychogenic nonepileptic seizure-like events (PNES). This finding is of critical importance for clinicians who deal with refractory cases of seizures, because misdiagnosis leads to administration of unnecessary antiepileptic medication with subsequent side effects and a significant financial burden (up to 4 billion USD).

PNES and epileptic seizure exhibit multiple overlapping clinical features. Ictal stuttering (IS) can be used as a useful sign to help distinguish between PNES from epileptic seizure in adult patients. In 2004, a study conducted by Vossler et al. compared two groups of patients with PNES and epileptic seizure, and evaluated them for IS. Interestingly, IS was observed only among patients with PNES (8.5% of 117 patients). Other features that assist clinicians to distinguish between these two groups of disorders are the “yellow” clinical characteristics of PNES. Specifically, seizures in PNES are usually characterized by gradual onset, a longer ictal duration (>2 min), and stimuli is often an emotional stressful event. Patients with PNES exhibit higher rates of psychiatric conditions, such as cluster A or B personality disorders, compared to patients with epileptic seizure.

EEG-video monitoring is the gold standard in PNES diagnosis. No EEG changes during a clinical event, accompanied by clinical spells inconsistent with seizure types that should induce changes in EEG recording, almost rule out PNES diagnosis [48,49,50].

Overall, given the diagnostic challenges posed by the clinical manifestations of both PNES and epileptic seizure, clinicians should be very careful, especially when evaluating refractory cases of seizures, because eventually misdiagnosis prevents patients from receiving suitable treatment for their condition.

## 10. Stuttering in a COVID-19 Patient

Finally, an interesting case of a patient who tested positive for SARS-CoV-2 and experienced newly developed stuttering and word-finding difficulties. Brain imaging tests were negative for acute pathology, and the patient was dismissed. Approximately ten days after her admission, her symptoms were only slightly improved. The etiology for the patient’s symptoms was unclear, and attributed to COVID-19 infection [13].

## 11. Conclusions

Stuttering is one of the commonly found symptoms or side-effects of hospital admitted patients with some sort of epilepsy. The diagnostic dilemmas clinicians face arise not only from the complexity of the phenomenon itself, but also from what scientific works reveal about stuttering, as these works inform the greater scientific community and in turn shape our understanding of the phenomenon. This paper has attempted to shed some light as to how the literature examines stuttering during the period 1960–2022. Our findings reveal our initial hypothesis on the disanalogous investigation of the topic holds true. In fact, 642 studies published on PubMed and Scopus investigate how epilepsy affects language. Interestingly, the number of publications on the topic drops to 50% when it comes to scientific articles on language-induced epilepsy, and only 62 articles engage in a study of how stuttering and epilepsy are connected. This observation, along with the connection of the neurological and psychogenic background of stuttering and epilepsy, comprise the clinicians’ diagnostic “bank” on cases that, as seen, show there is new ground in research to be covered. Reflex, induced stuttering, drug-induced stuttering, and COVID-19-related stuttering, along with genetic predisposition, are partly investigated, although interest is evident and case-studies beg for further research. This work reveals research trends and informs the scientific community of the pathophysiological and genetic background of stuttering and epilepsy, and draws attention to precise diagnosis. It was not yet possible to identify specific pathophysiological mechanisms shared by epilepsy and language as a trigger for ES. Stuttering is connected to many developmental and neurological syndromes and disorders, including Gilles de la Tourette syndrome, intellectual disability, obsessive-compulsive disorder, language impairments, and attention deficit hyperactivity disorder. The article reveals research trends and concludes that some sort of connections between stuttering and epileptic seizures has been the focus of studies/articles, and although conclusive findings that prove or rule-out direct connections/causality between stuttering and epileptic seizures are not yet at the disposal of the scientific community, attempts to investigate such connections by researchers are deemed important for the avoidance of misdiagnosis or overdiagnosis. Admittedly, the scientific community needs to invest more into the less proven mechanisms of stuttering and epilepsy to address diagnostic dilemmas.

## Figures and Tables

**Figure 1 jpm-12-00647-f001:**
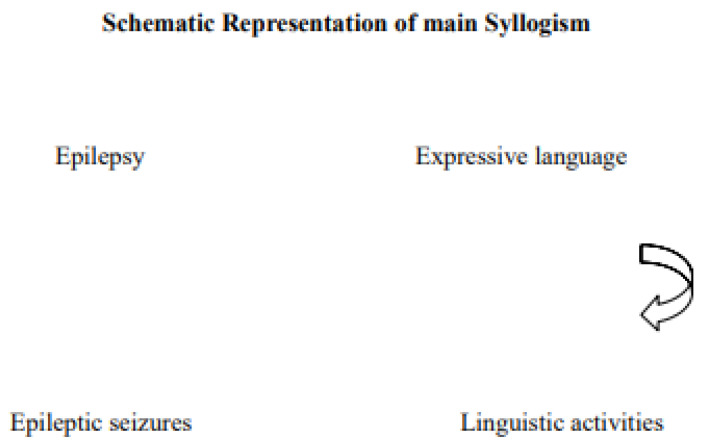
Schematic Representation of main Syllogism.

**Figure 2 jpm-12-00647-f002:**
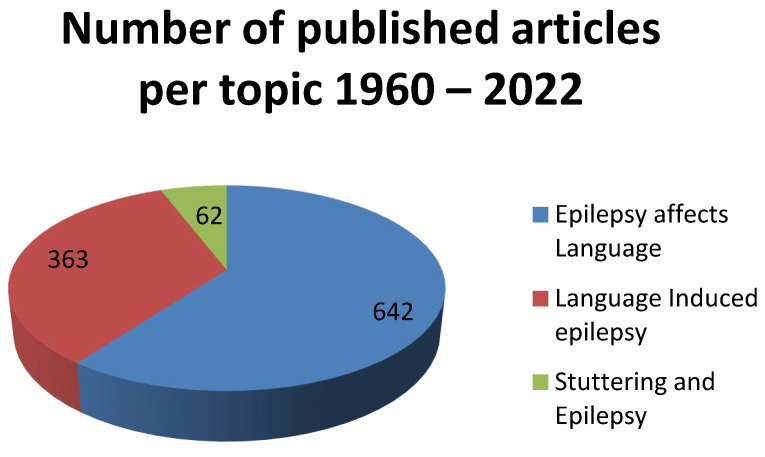
Number of published articles per topic 1960–2022.

**Table 1 jpm-12-00647-t001:** Literature Review Strategy: Number of published articles per topic.

Topic	Number of Published Articles
“Epilepsy affects language”	642
“Language-induced epilepsy”	363
“Stuttering and epilepsy”	62
Total number of particles	1067

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
