# Peer review of "Epilepsy and Diagnostic Dilemmas: The Role of Language and Speech-Related Seizures"

_jpm, 2022, doi:10.3390/jpm12040647_

Round 1

Reviewer 1 Report

"Epilepsy and Diagnostic Dilemmas: The Role of Language and Speech-related Seizures" is an interesting review article. The aim was to analyze the causal relationship between disorder-related language activities and epileptic seizures and literature review to examine the role of speech- fluency disturbances in the triggering epileptic seizures.

There are some issues in the article that need to be addressed.

1.- One of the main drawbacks of the manuscript is that the authors mention that there are 363 articles with the keywords “language-induced epilepsy” and only 62 with “stuttering and epilepsy”. However, due to the title and the objectives of the article, one would think that the review would focus on the 363 works, but this is not the case. The vast majority of the review focuses on articles on stuttering. In the paragraphs corresponding to "On language induced epilepsy" only 8 references of the 363 are used: References 3, 12, 21, 22, 23, 24, 25 and 26. It is suggested to make an adequate selection of the 363 relevant articles due to their solidity. methodological, describe what criteria were used for said selection and on said selected articles make a broad and deep review with a critical analysis of each of the topics discussed. One of the topics to be discussed would obviously be stuttering.

2.- Another aspect that is suggested to be improved is the coherence between the referred articles. In the current form, a disconnection between the paragraphs or the works mentioned is noted in various segments of the manuscript, for example: In the first paragraph of the section “The neurological and psychogenic background of stuttering and epilepsy” a disconnection between the two statements is perceived.

3.- Finally, it is suggested to improve the depth with which the pathophysiological mechanisms involved in language as a trigger of epileptic seizures are described. At the time of mentioning an article, the most relevant findings and the pathophysiological explanation of these results should be delved into. For example on page 11, when citing references 40 and 41 “Nevertheless, pharmacological studies have implicated dopamine [40, 41]”. Another example is in the last paragraph of the section "The role of stuttering and connection with reflex-epilepsy" Video-EEG studies are mentioned but the results in the recording of the paroxysmal activity of said patients are not described. By the way, there is no reference to this study. And the description of more articles that use current techniques such as EEG, Video-EEG, PET, fMRI, etc. that support the pathophysiological mechanisms that occur in language and epilepsy is missed.

The previously mentioned drawbacks merit a careful review and rewriting of the manuscript in order to improve its quality.

Author Response

This article is to serve as a precursor for a systematic literature review of all article reviews in the field in the near future, which is why for the time being an addition in the method section clarifies under which grounds the specific number of articles has been chosen. More specifically:

To address disconnections between paragraphs and works mentioned, the manuscript’s section “The neurological and psychogenic background of stuttering and epilepsy” was rewritten.

We presented (chronologically) the different approaches regarding the pathophysiology of stuttering and the extensive use of modern neuroimaging techniques used, explained the possible implication of neurotransmitters (e.g., Dopamine) in the pathophysiological mechanism of stuttering, and offered insight into the limited findings on its genetic background and the possible genetic relationship between epilepsy and stuttering. Afterwards, we introduced language as a trigger to epileptic seizures using case reports and a family study. In the chosen research articles, patients were studied using video electroencephalographic recordings with EMG, whose ictal findings we mention in our main text. Given the current bibliography, it was not possible to establish a pathophysiological mechanism shared by epilepsy and language as a triggering factor for ES.

Reviewer 2 Report

stuttering is often considered a psychiatric / psychological phenomenon which is why many neurologists do not study it extensively or factor it into their decision making. many studies have been performed to look at the associations between language and epilepsy using a variety of techniques - IAP, MEG, F-MRI, etc. describing such studies would be of great value in expounding the relationship between epilepsy and language.

Author Response

In a separate section of our manuscript, the diagnostic dilemma between psychogenic nonepileptic seizures and epileptic seizures is highlighted. We present the possible role of stuttering in diagnosing either condition, and cite evidence that assist clinicians in making proper diagnosis.

Because it was not possible to establish a pathophysiological mechanism shared by epilepsy and language as a triggering factor for ES (a very particular relationship), given the current bibliography, we decided to present the pathophysiological mechanisms of stuttering through findings of modern neuroimaging techniques (MEG, MRI, fMRI, PET). Subsequently, we conclude that there is ground for research on the aforementioned field.

Round 2

Reviewer 1 Report

The authors have made the suggested corrections. There are still some errors in the spelling of some words, for example in the 9th line of the abstract “explorespathophysiological andthe” the words are joined. It is suggested to carefully review the entire manuscript in search of this type of error. 

Author Response

REPLY TO 1st REVIEWER

We would like to express our gratitude to the reviewer for the positive impact. His comments from the first round of the review helped us to significantly improve the quality of our manuscript.

In terms of the errors in the spelling of some words, I am afraid that this is purely related to the format, considering that in our original word document, there are not any joined words. I am sure that the editor will kindly check that in the upcoming steps.

Reviewer 2 Report

the paper moves in different directions with regards to the role of language in epilepsy. it may help to focus on a single aspect of language and epilepsy rather than focus on multiple aspects within the same paper. some topics like stuttering are not strongly linked to epilepsy.

Author Response

REPLY TO 2nd REVIEWER

We would like to take the opportunity to express our gratitude to the reviewer for his kindness to provide us with his constructive comments.

The article looks at literature and attempts to draw connections. Since our literature review reveals attempts to connect stuttering with epilepsy, in any way, these attempts cannot but be included in our article.

The part titled: "The role of stuttering and connection with reflex-epilepsy" attempts to show attempts to link stuttering with epilepsy.

We have references that indicate possible connection. (i.e. "As indicated in previous sections of the current review and by relevant bibliography, the rates of stuttering among patients with epilepsy are higher than in the general population. The interrelation between epilepsy and stuttering is not straightforward, especially when stuttering is considered the stimuli for an epileptic seizure to occur, and not the clinical symptom of an ES [29, 30, 43]."

Also: "From a genetic perspective, intragenic deletions of the contactin-associated protein-like 2 gene (CNTNAP2) have been found in patients with epileptic syndrome and stuttering"

And: "Despite the limited available bibliography on ES induced by language, there have been case reports and studies that used video-EEG with electromyogram (EMG) recordings in an attempt investigate stuttering patients. In a case of language-induced epilepsy, the patient exhibited facial myoclonus, while reading aloud, and dysfluent language, mimicking stuttering. Paroxysmal discharges in EEG recordings of the left frontal region were consistently associated with a brief interruption of language. In this particular patient, silent reading did not induce any epileptic discharge, and as a result, articulatory movements during phonation were assumed to be the triggering factor. The patient was treated with antiepileptic medication [29]."

Similarly: "Michel et al conducted video-polygraphic EEG recordings in four patients with a diagnosis of juvenile myoclonic epilepsy (JME) in whom coexisted praxis- and language-induced jerks. Complex stimuli, reading and praxis, induced reflex seizures in these patients, characterized by facial myoclonias and stuttering. EEG recordings were indicative of brief paroxysms of very fast spikes followed by a slow wave, mainly in the frontocentroparietal areas [27]."

Likewise: "In a case of language-induced epilepsy, the patient exhibited facial myoclonus, while reading aloud, and dysfluent language, mimicking stuttering. Paroxysmal discharges in EEG recordings of the left frontal region were consistently associated with a brief interruption of language. In this particular patient, silent reading did not induce any epileptic discharge, and as a result, articulatory movements during phonation were assumed to be the triggering factor. The patient was treated with antiepileptic medication [29]. Michel et al conducted video-polygraphic EEG recordings in four patients with a diagnosis of juvenile myoclonic epilepsy (JME) in whom coexisted praxis- and language-induced jerks. Complex stimuli, reading and praxis, induced reflex seizures in these patients, characterized by facial myoclonias and stuttering. EEG recordings were indicative of brief paroxysms of very fast spikes followed by a slow wave, mainly in the frontocentroparietal areas [27]. In a neurophysiological study of nine members of a family with history of idiopathic generalized epilepsy (IGE) with interictal stuttering, spontaneous language was the main triggering factor for the occurrence of myoclonic jerks in five members. They also reported acquired stuttering (which was proved to be of epileptogenic origin). Stuttering was also present while reading and EEG recordings exhibited abnormalities (e.g., spikes followed by slow waves) [21]. The above observations suggest that some forms of acquired stuttering could be linked to ES and electrophysiological studies may prove useful in investigating them."

According to the 2nd reviewer's comment, we have added the following in the conclusion of our paper: "The article reveals research trends and concludes that some sort of connections between stuttering and epileptic seizures has been the focus of studies/articles. And although conclusive findings that prove or rule-out direct connections / causality between stuttering and epileptic seizures are not at the disposal of the scientific community, yet, attempts to investigate such connections by researchers are deemed important for the avoidance of misdiagnosis or overdiagnosis. Admittedly, the scientific community needs to invest more into the less proven mechanisms of stuttering and epilepsy to address diagnostic dilemmas".